# Stress Management in Healthcare Organizations: The Nigerian Context

**DOI:** 10.3390/healthcare11212815

**Published:** 2023-10-24

**Authors:** Ezinne Precious Nwobodo, Birute Strukcinskiene, Arturas Razbadauskas, Rasa Grigoliene, Cesar Agostinis-Sobrinho

**Affiliations:** 1Faculty of Health Sciences, Klaipeda University, LT-92294 Klaipeda, Lithuania; ezinne.precious.nwobodo@ku.lt (E.P.N.); birute.strukcinskiene@ku.lt (B.S.); arturas.razbadauskas@ku.lt (A.R.); 2Faculty of Marine Technologies and Natural Sciences, Klaipeda University, LT-92294 Klaipeda, Lithuania; rasa.grigoliene@ku.lt

**Keywords:** stress, stress management, healthcare, healthcare professionals, Nigeria

## Abstract

Occupational psychosocial stress can increase the risk of several cardiometabolic diseases. Healthcare workers worldwide experience exceptionally high levels of occupational stress, leading to serious individual, organizational, and societal problems. This narrative review seeks to provide information about the overall consequences of having over-stressed healthcare workers and ascertain how it eventually holds back the advancement of healthcare. In addition, we present a review of the concept, study, and theories related to stress management in order to deeply understand this issue, providing a theoretical perspective of stress management and the subtle concepts of stress, stress management, healthcare structure, and organization in Nigeria. The current literature has shown that Nigerian healthcare workers are more stressed due to long working hours, caregiving responsibilities, and psychological contact with patients. Healthcare workers are more likely to experience stress and burnout than other professions. The level of stress in the healthcare sector has garnered a lot of attention in this regard because of the negative impact of stress on both staff and patients. However, health policies and better working conditions need to be adopted. Collaborative efforts from policymakers, healthcare institutions, and other stakeholders are necessary to prioritize the well-being and productivity of healthcare professionals in the journey toward a more robust and equitable healthcare system.

## 1. Introduction

Healthcare professionals play an indispensable role in society today, providing healthcare services to improve health outcomes. Health professionals are the backbone of the social and healthcare management force since they are responsible for promoting health, preventing the spread of disease, and providing healthcare services to children, adults, families, and communities in accordance with the primary healthcare strategy [1]. The global advent of the novel coronavirus in 2020 further showed the crucial nature of healthcare providers and professionals in driving public health objectives [2]. However, based on the intensive care that healthcare professionals provide inside and outside health facilities, they are often exposed to physical and emotional stress, which can inhibit their productivity and performance if not well-managed [3,4]. One in three employees worldwide experiences work-related stress [5]. Moreover, stress at work has been linked to health ailments like sleeplessness, depression, and anxiety [6,7,8,9].

Moreover, all professions experience stress at work, but because of the nature of their work environment, healthcare professionals often face severe stress daily [10,11]. Healthcare professionals frequently have to deal with some of the most difficult situations in any profession. In [7], the authors emphasized that healthcare professionals come into contact with quite a number of patients, all with different health problems; these health issues may be life-threatening and difficult to treat due to their obscurity and uncertainty, causing stress to arise in the patient as well as in the personnel in charge of monitoring their health status. According to the American Psychological Association (APA), the three main/primary types of stress that a person can experience are acute, episodic acute, and chronic stress [12]. For example, the author states that acute stress emerges from our body’s reaction to a novel or challenging situation; it happens very quickly and can throw the individual off balance [13]. Episodic acute stress refers to the re-occurrence of acute stress at close intervals. Moreover, chronic stress occurs and persists over an extended period of time, causing physiological and physical health implications, as well as burnout [14].

Stress experienced by healthcare professionals substantially impacts behavioral changes, which can cause people to quit their jobs, change jobs, or end their relationships with co-workers [15]. This effect of stress on healthcare professionals, if not well-managed, can expose the health sector to a chronic crisis. Therefore, stress management is crucial for healthcare professionals to cope with workplace stress in order to improve work performance. In this regard, stress management means adopting a coping mechanism to control stress and remain productive at work [16].

The central intention of stress management is to assist individuals in handling likely stressors that may affect them and the emotional, physiological, and health consequences that emerge as a result of the aforementioned stressors by attempting to switch their emotional responses to the stressors. Stress management requires a coping mechanism that can be psychologically, physically, and environmentally driven in order to manage workplace stress among healthcare workers [17].

Healthcare professionals working in Nigerian healthcare facilities are significantly overwhelmed with work and daily tasks [18]. Prior to the pandemic, data have shown that in the European Union, psychosocial stress affected around 22% of workers; in Nigeria, a recent study showed that, overall, the prevalence of psychosocial stress was 61.97% [19]. Most employee stress in Nigerian hospitals is brought on by a work overload, monotonous or repetitive tasks, a lack of resources, an unfavorable physical or psychological work environment (such as verbal abuse or inappropriate behavior), working long hours without taking annual leave or lunch breaks, problems with people management, an insufficient division of tasks, and the use of primitive technology [19,20]. Similarly, the stress experienced by healthcare professionals in Nigerian hospitals is exacerbated by the nationwide shortage and emigration of healthcare professionals. For instance, the authors of one study stated that Nigeria’s doctor–patient ratio is 1:6400, which is much lower than the advised ratio of 1:600 [21]. This indicates that the few available doctors are severely stressed, taking care of the health needs of about 200 million people [22].

However, taking all of this into account, necessary information collected in a single document is needed to better understand the answer to the following question: “What is the current situation of stress management in the context of Nigerian healthcare organizations?” Healthcare professionals undergo severe stress and require effective management. Hence, the main aim of this review is to provide current general knowledge on stress management as well as on how it is carried out in Nigeria’s health sector.

## 2. Methods

With this review, we aimed to search for broad inclusion and search criteria to map the available evidence, identify the evidence’s important features, and identify current knowledge gaps. Similarly, it serves as a precursor for future and more accurate and valuable studies on stress management in healthcare organizations. A narrative review was performed, conducting a comprehensive literature search of key electronic databases (Scopus, Pubmed—MEDLINE) and search engines (Google Scholar) for peer-reviewed publications and gray literature (Klaipeda, Lithuania). The literature search was not restricted by timeline or language. Articles were evaluated by title, abstract, and full text. After that, duplication between conference abstracts and journal articles was checked. In the case of duplication, only the full text was chosen. Search terms were used to comb through numerous databases to find pertinent research for review (Table 1). The search technique used in this study complies with Higgins and Green’s (2011) demands for a search to be effective, transparent, repeatable, and resilient.

### Theoretical Framework

This study’s theoretical approach is based on two key models: the job demands–resources (JD-R) model and multidimensional health theory (MHT) [23,24].

The job demands–resources (JD-R) model [23] provides a comprehensive framework to understand workplace stress and its impact on employee well-being and engagement. According to the JD-R model, job resources, such as autonomy, feedback, and supervisor support, play a crucial role in enhancing work engagement. These resources provide employees with the necessary tools and support to effectively perform their tasks, leading to higher motivation and job satisfaction. Additionally, personal resources, such as self-efficacy, optimism, and resilience, also contribute to work engagement and serve as buffers against the negative effects of job demands. On the other hand, the JD-R model posits that job demands, such as role conflict, role ambiguity, and emotional expectations, can drain energy and lead to adverse employee outcomes, like strain and burnout [23]. These job demands can vary depending on the occupation, making the level of stress experienced in one profession differ from another. The JD-R model’s strength lies in its ability to explain the complex interplay between job resources, job demands, and employee well-being, highlighting the importance of maintaining a balance between these factors to promote a healthy work environment; as such, this makes the level of stress experienced in one occupation vary from another. Interestingly, the correlations proposed using JD-R models have received support from numerous cross-sectional, meta-analytic, and multilevel investigations [25,26,27,28].

On the other hand, multidimensional health theory (MHT) [24] provides a comprehensive understanding of psychological health, encompassing both cognitive and physical dimensions. The theory proposes two specific hypotheses: First, it suggests an inverted-U link between somatic stress, depression, anxiety, and health. This means that up to a certain point, these factors may enhance cognitive performance and health, but beyond that point, they start to negatively impact an individual’s well-being. Second, MHT hypothesizes a negative linear relationship among cognitive states, such as stress, depression, anxiety, and health. As cognitive stressors increase, an individual’s cognitive performance and overall health decline [24]. MHT emphasizes how workplace stress can influence individual performance not only in work settings but also in social settings. For example, when individuals experience heightened cognitive stress, anxiety, and depression, their performance at work and overall health may deteriorate. Similarly, in somatic conditions, such as athletic performance, a certain level of anxiety, stress, and depression may actually enhance performance, but beyond a certain point, it becomes detrimental to athletic achievements.

It is evident from both the JD-R model and MHT that workplace stress significantly affects employee performance and health status [29]. As a result, it is crucial to implement effective stress management strategies to enable workers to cope with occupational stress. By promoting a supportive work environment and providing resources to address job demands, organizations can enhance employee well-being, productivity, and overall job satisfaction. Additionally, understanding the complexities of workplace stress using the JD-R model and MHT can guide the development of targeted interventions to promote the mental and physical health of healthcare professionals and other workers in various industries.

## 3. The Concept of Stress and Occupational Stress

Stress is described as a severe, exacting, or difficult occurrence or scenario. Stress can also be defined as a physical, mental, or emotional component that generates physiological or mental strain in a medical or biological environment. According to Arora et al. [30], stress has been related to the onset of various illnesses that contribute significantly to the societal health burden and cause significant morbidity and death in people of all ages. In the view of Berhanu et al. [31], stress is a state of emotional or physical stress. Any circumstance or idea that gives an individual a cause for annoyance, rage, or anxiety can trigger stress [32].

Furthermore, stress is the body’s response to a demand or difficulty. Stress can be beneficial for certain periods of time, such as when it occurs to avoid danger or meet a deadline. However, prolonged stress might have a negative impact on one’s health [33]. As emphasized by Berhanu et al. [31], there are two main categories of stress that can be established: eustress (positive stress) and distress. People who encounter eustress will be able to satisfy their obligations at work, which may help them have a more fulfilling work life (for example, satisfaction and positive moral values) [3]. On the other hand, people in distress may be motivated to reduce the quality of their working lives because they cannot meet the demands of their jobs (for example, unhappiness and poor moral principles) [34]. However, the consequences of continuous stress are enormous. This affects individuals’ psychology, health, work productivity, and social life [35].

Further, one in three employees worldwide is thought to experience work-related stress [15]. All professions experience workplace stress, but because of the nature of their work environment, healthcare professionals represent a significant population that can be impacted [36]. In this regard, the Centers for Disease Control and Prevention (CDC) [37] defined occupational or workplace stress as a dangerous physical and emotional reaction that occurs when the job requirements do not meet the worker’s capabilities, resources, and demands. Hence, work-related stress and sadness are characterized as negative reactions that people have to exceed expectations and demands at work. According to the most recent data from the Labor Force Survey (LFS), in 2020–2021, there were 822,000 work-related stress, depression, or anxiety incidents, with a prevalence rate of 2480 per 100,000 employees. In comparison to the previous year, this rate is not statistically significant. Before the coronavirus pandemic, self-reported work-related stress, sadness, and anxiety rates had increased in recent years. The rate in 2020–2021 was greater than pre-coronavirus levels in 2018–2019 [38]. Therefore, stress and the workplace stress discourse have occupied the central argument in the literature and how it can be managed to improve workplace productivity, especially in health organizations.

### 3.1. Types of Stress

According to the American Psychological Association (APA), the three main/primary types of stress that a person can experience are acute, episodic acute, and chronic stress [12]. However, each of the three categories of stress has unique traits, symptoms, time frames, and methods of therapy. Therefore, a detailed explanation is given below.

### 3.2. Acute Stress

Acute stress emerges from our body’s reaction to a novel or challenging situation; it happens very quickly and can throw the individual off balance. It surfaces when something perceived as a threat emerges or is unexpected, and it leads to bouts of anxiety, irritability, headaches, stomach upsets, rapid heartbeats, and sweating [39]. With the presence of acute stress, the brain signals the release of a host of stress hormones like cortisol into the bloodstream. These hormones, in turn, trigger the fight-or-flight mechanism in the body, including but not limited to, an increased heart rate, respiration, the breakdown of carbohydrates and fats, and an increase or a decrease in blood pressure (physical stress). The quickening of these physical processes also slows down certain physiologies of the body, such as the flow of blood to the digestive system, severely debilitating appetite, and the need for food intake (physiological stress).

### 3.3. Episodic Acute Stress

This refers to the re-occurrence of acute stress at close intervals. It may surface as a result of taking on too much responsibility; the individual experiencing episodic acute stress may suffer from severe burnout [40]. It is prevalent in people who set unrealistic goals or demands for themselves, causing them stress in their attempt to try to achieve those goals. Symptoms of episodic acute stress disorder include pain attacks, heartburn, uncontrollable anger or irritability, and unintended hostility (behavioral stress), and if left untreated or unattended to, episodic acute stress can lead to heart disease, hypertension, and frequent migraines. This kind of stress appears to take a toll on the mental well-being (cognitive and mental stress) of the person(s) suffering from it [41]. Although this particular kind of stress is fleeting, a whole range of physical health problems is associated with joint pain, low blood pressure, cardiovascular disease, and so on. Anger, frustration, and depression (psychological and emotional stress) are also common antecedents of this type of stress.

### 3.4. Chronic Stress

According to Chandola et al. [14], chronic stress occurs and persists over an extended period of time. It is the most damaging kind of stress to our health [39]. This kind of stress causes physiological and physical health implications as well as burnout. Similar to episodic acute stress, this type may cause joint pain, an increase or a decrease in blood pressure, and cardiovascular disease (physical and physiological stress). Chronic stress is prolonged and continual stress that has severe effects on the immune, neuroendocrine, central nervous, and cardiovascular systems [42,43]. One of the major causes of chronic stress is challenges and pressures from our jobs and work environments, as well as relationship stress, i.e., strained relationships between friends, co-workers, partners, and family.

### 3.5. Stress Management

Kema et al. suggest that a remedy can only be discovered after understanding the nature and origin of stress [44]. Hence, understanding how stress affects performance will help individuals to cope with and manage stress [45]. When an individual is self-conscious and aware of the symptoms of stress, one can use it constructively rather than destructively [46]. Therefore, for stress management, it is important to recognize when people are approaching a destructive and unproductive condition. Similarly, Non et al. [47] suggested that the first step in managing stress is to become more aware of yourself (know thyself), including how you react in various situations, what stresses you out, and how you behave when you are stressed. Adjusting your lifestyle if continually stressed, preventing stress with self-care and relaxation, and controlling your response to stressful situations when they emerge are all examples of stress management [48].

In this respect, stress management is the utilization of a wide range of treatments and psychotherapies to help people control their level of stress, especially chronic stress, usually with the goal of enhancing daily functioning [18,49]. According to Tawfik et al. [50], stress management refers to a set of techniques for dealing with stress and difficulties in one’s life. More so, stress management is an intervention meant to lessen the effect of stressors at work [51]. However, these interventions may be personalized with the goal of enhancing a person’s capacity to handle pressure [52]. Moreover, the main goal of stress management is to protect people’s mental and physical health, quality of life, and daily productivity [33]. It is crucial to recognize the stress that directly influences the success of healthcare facilities, to be aware of its signs, and to know how to manage it in order to reduce the harmful effects that it will have [53]. The key to stress management is to know the difference between the proper amount of stress, which gives one energy, ambition, and excitement, and the wrong amount of stress, which can ruin one’s health, outlook, relationships, and well-being [54].

Additionally, stress management strategies can be split into two categories [55]. In this regard, sustainability management comes first. Sustainability management should create conditions at work that reduce sources of stress, as well as take steps to help employees deal with a range of stressful situations effectively [55,56]. The second approach is the personnel management approach. Employees are at the center of the personnel stress management strategy. This method enables staff to create a daily to-do list, prioritize items on the list, and schedule tasks accordingly [55,57]. This approach makes workers relax while working, take regular pauses, and manage their time well so that employees can meet deadlines, deal with pressure at work, and minimize stress [58].

### 3.6. Drivers of Stress in Healthcare Organizations

According to Trifunovic et al. [59], individual factors such as physical health, the quality of interpersonal relationships, the number of commitments and responsibilities of employees, the degree of others’ dependence on workers and work expectations, the level of support, and the number of changes or traumatic events that have recently occurred in workers’ life all influence the level of stress experienced at work, especially in healthcare organizations. In the opinion of Melaku et al. [60], healthcare professionals often face stress at work due to workload demands, which are always greater than their capacity to handle them. Similarly, Cleary et al. [61] suggested that healthcare professionals work in some of the most demanding conditions, which makes them more exposed to stressors than other occupations.

Additionally, the advent of COVID-19 put healthcare workers under even more strain, necessitating the development of new stress-relieving techniques. Odigie [62] postulated that healthcare professionals are subjected to various job stressors that can negatively impact their mental and physical health and their productivity at work. Healthcare professionals confront stress every day because of their work environment [63]. Due to the work environment of healthcare professionals, Odigie [62] pointed out that long working hours, ineffective management, personal life, interpersonal relationships, organization factors, and the work environment are some of the inherent factors that have led to the classification of workplace stressors commonly encountered by healthcare professionals in the healthcare environment. Hence, these identified factors are further explained below.

### 3.7. Long Working Hours

As postulated by Wong et al. [64], the issue of working long hours in an organization has been a contentious topic since the 1980s, when a Japanese design engineer died of a brain hemorrhage after working 2600 h a year. This has sparked various research on the influence of long working hours on the health of workers in various industries for various health reasons. Work overload is common in most organizations and businesses, where the amount of time spent on work, which includes core activities, associated jobs, commuting, and travel, is excessive and is actively or passively harmful to workers’ health. In the opinion of Caruso [65], long work hours increase the risk of short sleep duration and sleep disruption, and this is mostly observable in healthcare institutions, as workers are typically required to offer patient care around the clock. Thirty-two percent of healthcare professionals attest to being sleep deprived due to long work hours, which, especially among them, can lead to burnout [65,66].

According to Clark et al. [67], long hours of work without sleep can affect the circadian rhythm. In this regard, the circadian rhythm refers to a person’s normal sleep-wake cycle corresponding to the time of day. The lack of sunshine causes the brain to release melatonin when individuals operate at night [68]. This hormone has a sleep-inducing effect on the body. Therefore, healthcare professionals who work the night shift upset their biological cycle, resulting in extreme stress [67]. Moreover, mental weariness can be triggered as a result of this interruption, and this exhaustion also causes stress.

However, long working hours mostly occur in healthcare organizations as healthcare professionals seek to enhance patient safety while lowering the risk of harm [69]. Despite healthcare professionals’ daily efforts to save lives, the rate of medical mistakes continues to soar, resulting in considerable impairment and death due to the stress they encounter on a daily basis [70]. In the same manner, according to West et al. [71], if medical errors are linked to fatigue and anxiety caused by long working hours, then healthcare professionals who are overworked are more prone to making mistakes and providing poor healthcare [69]. This is due to stress and, as a result, jeopardizes patient safety [72]. Some of the medical errors that can occur in healthcare systems due to stress at work include medication errors, anesthesia errors, hospital-acquired infections, missed or delayed diagnosis, avoidable treatment delays, inadequate post-treatment follow-up, inadequate monitoring after a procedure, failure to act on test results, failure to take proper precautions, and technical medical errors [70,72].

### 3.8. Ineffective Management

Poor management of staff creates an unstable and toxic work environment and atmosphere, reducing efficiency and adversely affecting the quality of life of healthcare professionals [73]. Moreover, long hours, job overload, time constraints, challenging or complex duties, a lack of breaks, a lack of variety, and unfavorable physical working conditions are all indicators of inadequate management [74]. This is because poor management results from bad and egocentric leadership skills. In addition to this, ineffective management is a source of emotional stress at the workplace; thus, nurses with superiors who are not leaders exacerbate the stress that the subordinates go through. For instance, Karyotaki et al. [75] noted that poor managers could significantly harm both staff productivity and health. This implies that good management may unite people into a team and foster their commitment to a business, but bad management can distract employees and foster a toxic work environment, thereby instigating stress.

### 3.9. Personal Life and Interpersonal Relationships

The fact that healthcare professionals have personal lives apart from the workplace raises the possibility of stress. This is a result of attempting to strike a work–life balance. As emphasized by Moustaka et al. [76], work relationships also act as potential stressors; disagreements and conflicts among co-workers, and a lack of social support between staff, colleagues, and superiors, also serve as significant stressors, as a work environment where distrust, lack of camaraderie, and friendship are prevalent would not accommodate the smooth running of the business. Healthcare is far too important to be bogged down by toxic relations among personnel, as patients’ lives depend on effective and efficient teamwork.

### 3.10. Organizational Factors

Studies have shown that coupled with the laborious tasks of healthcare jobs, organizational and managerial characteristics directly influence the level of stress healthcare professionals experience in the workplace [76]. More so, Konstantinos and Haque et al. [77,78] showed that a huge percentage of possible stressors among healthcare professionals relate to the nature of the space in which they work: having to attend to patients’ physical and psychological demands, keeping up with what job demands, inter-hospital competition, the lack of autonomy in carrying out tasks, and so on, may lead to emotional as well as physical exhaustion. When there is a lack of standard organizational structure, chaos ensues.

### 3.11. The Work Environment

Stress caused by bad working environments can be extremely disconcerting and uncomfortable, especially in healthcare institutions. The organizational culture may also play a significant role here [74] including a poor physical environment, pressures of time, a heavy workload, role conflict, role ambiguity, poor or bad relations with bosses, superiors, subordinates, or background, finance, and social support factors. If the workplace is not convenient for efficient work to be completed, there is a high likelihood of errors, mistakes, mix-ups, and mishaps [79]. However, role ambiguity (a lack of clarity regarding roles, responsibilities, or expectations), role overload (perception of excessive tasks, responsibilities, or demands in a given time), and role conflict (contrasting demands or priorities within one or more roles) were identified as the three main drivers of stress in the work environment by [80]. In this regard, the work environment can constitute a major source of stress for healthcare professionals if not well-organized.

### 3.12. The Stress Management Approach

Understanding how stress affects one’s performance is essential for managing stress. Being self-aware and able to identify the signs of stress enables us to channel it positively and avoid its negative effects. However, no particular approach can be adopted to manage stress because stressors vary; however, Nahavandi et al. [81] identified two broad approaches to stress management in healthcare organizations. The first is the personal strategy and the second is the organizational strategy of stress management. Figure 1 demonstrates the approaches suggested by Nahavandi et al. [81].

According to Figure 1, personal strategies involve lifestyle modification, attitude adjustment, social support, emotional regulation, and time management to effectively cope with stress. Health is crucial for dealing with demands and problems, and personal choices impact physical health and well-being. Sindhu et al. [82] noted that a personal approach can be adopted in the workplace to cope with stress. This has to do with the adjustment of how individuals react or respond to work situations. In order to maintain excellent health, it is crucial to consume a balanced diet, get adequate sleep, and engage in regular exercise. These actions aid in preventing some of the negative effects of stress and in coping with its symptoms. Coping with stress in an organization, therefore, necessitates the universal acceptance of this personal approach. However, researchers [83] stressed the role of time management as a personal approach to stress management, where managing time is handling stress. This is because good time management enhances one’s ability to set and review priorities, keep track of time spent, and plan their time to better reflect their obligations, plans, and goals. This approach helps a person to be organized and reduce the stress encountered during the execution of plans.

Furthermore, the second approach shown in Figure 1 is the organizational approach to stress management. Nahavandi et al. [81] stated that the most important step an organization can take to alleviate stress is to redesign the role of the job and support workers. This is because some occupations are stress-oriented by nature; hence, modifying them to match employees’ demands and lessen some causes of work-related stress is essential [84]. Furthermore, the provision of incentives, flexible work plans, and health insurance packages can be an appropriate strategy that can complement organizational efforts in handling workplace stress, especially among healthcare workers.

## 4. Nigeria’s Healthcare Context

Nigeria finds itself at a critical juncture, grappling with the challenges posed by a rapidly growing population of approximately 200 million people [84,85]. In light of this demographic surge, Nigeria is facing significant obstacles in effectively addressing the population increase and climate vulnerability, leading to shortcomings in meeting MDGs (Millennium Development Goals) and experiencing sluggish progress toward SDG (Sustainable Development Goal) targets [84]. Disturbingly, health indices depict dismal outcomes and a concerning decline in health conditions spanning the past three decades. This unfortunate situation can be attributed to the lack of consistent access to high-quality healthcare, education, and essential public services. This inadequacy, in turn, has the potential to escalate societal unrest, drive large-scale unplanned migration, and ultimately lead to regional and even universal destabilization [84]. Nigeria’s primary, secondary, and tertiary health service delivery systems are functioning inequitably, prompting patients to avoid utilizing primary healthcare (PHC) facilities when such services lack trustworthiness, affordability, or acceptable quality [86]. An underlying factor contributing to the system’s failure lies in the uneven distribution of responsibilities between federal, state, and local government levels, which places undue strain on tertiary institutions operating in regions with inadequate secondary care [87]. The funding challenges faced by state governments in Nigeria further exacerbate the situation, resulting in an insufficient number of healthcare personnel and, consequently, leading to disparities and brain drain, further compounded by significant emigration.

To alleviate stress and enhance the quality of healthcare services, a transformative shift in government policy is imperative. This includes an increased allocation of funds, bolstering the healthcare workforce, improving infrastructure and facilities, and embracing the digitization of healthcare practices. Abubakar et al. [84] also emphasized that the government could increase health spending by dedicating a budget that is equitably allocated at both federal and state levels, ensuring justice and transparency with public and independent audits.

In conclusion, rectifying the pressing issues within the Nigerian healthcare system requires a comprehensive and coordinated effort, guided by evidence-based policies and substantial investment. By prioritizing healthcare, Nigeria can pave the way for an improved and resilient healthcare system that can effectively meet the needs of its growing population and contribute to the nation’s overall development and stability.

### 4.1. The Structure of Healthcare Organization in Nigeria

In Nigeria, the healthcare system is organized according to the levels of the government. The local government is in charge of primary healthcare, the state is in charge of secondary healthcare, and the federal government oversees all three [88]. Through the National Health Bill, the federal government establishes requirements and controls how both commercial and public health providers in Nigeria perform healthcare services [87]. The structure of healthcare organizations in Nigeria is composed in Primary Health Care; Secondary Health Care and Tertiary Health Care [87].
Primary Healthcare (PHC) Organizations: PHC is a community-based healthcare center that the state and federal governments control. Nigeria’s Federal Ministry of Health (FMOH) has emphasized that PHC provides less intrusive minor diagnostics and essential medical treatments. PHC serves as an individual, family, or community’s initial point of contact with the national health system [88]. Furthermore, the World Health Organization emphasizes that Nigeria’s primary healthcare system is the most pertinent, distinctive, and important part of the nation’s three-tier healthcare system [89]. Furthermore, due to its proximity to patients compared with other types of healthcare, primary healthcare is better able to meet the local population’s demands for healthcare. Most importantly, Koce et al. [90] pointed out that everyone initially uses the primary healthcare system because it is accessible and designated as the first line of defense in the event of a medical emergency, regardless of social or economic background. Additionally, PHC occupies the lowest position in Nigeria’s healthcare system. A total of 88% of Nigeria’s healthcare facilities, according to Makinde et al. [91], are PHC facilities.Secondary Healthcare (SHC) Organizations: “General Hospitals” is another name for this group. The state government oversees s SHC’s business through regulation and oversight. This is due to the fact that healthcare providers have a higher status than PHC and are thought to provide patients with more complex healthcare services [88]. In reality, Balogun [87] noted that PHC providers refer complex and difficult-to-manage medical problems to SHC providers. SHC provides other community health services in addition to general medical, surgical, pediatric, obstetrics, and gynecological care [91]. Private healthcare delivery services are included in Nigeria’s healthcare service delivery category. In contrast to PHC, SHC has considerable staffing and facility capacities.Tertiary Healthcare (THC) Organizations: THC includes the top healthcare organizations in Nigeria’s healthcare system. This cadre of the country’s health system delivery includes teaching hospitals, federal medical centers, and other specialized hospitals [85]. More specifically, THC serves as a hub for medical research, as the name suggests. The federal government is responsible for regulating and controlling THC. Around 0.25% of all the country’s medical facilities are in THC. The organizations’ control and management are exclusively reserved for the federal government [88].

Furthermore, the structure of healthcare organizations paves the way for the private sector to participate in the provision of healthcare in the nation to support secondary healthcare systems [92]. With barely 5% of the budget going toward healthcare, Nigeria’s healthcare system is plagued by a lack of financing and investment [85]. The WHO ranked Nigeria as low as 156th out of 191 countries for the caliber of its healthcare delivery system [89]. This poor performance of Nigerian healthcare organizations has exacerbated the stress that health professionals go through daily in their workplaces. Hence, there is a need to analyze the stress management mechanism in healthcare organizations in Nigeria in order to suggest policies that enhance stress management and improve the work productivity of healthcare workers in the region.

In Nigeria, these three tiers of healthcare organizations, namely, primary, secondary, and tertiary healthcare systems, contribute to varying stress levels experienced by healthcare professionals. At the primary level, where basic medical services are provided, healthcare workers often face stress due to limited resources, overwhelming patient volumes, and challenging working conditions. In the secondary level, stress arises from the increased complexity of cases and higher patient expectations. Additionally, healthcare professionals at this level may also deal with inter-professional conflicts and administrative pressures. Finally, at the tertiary level, where specialized and critical care is delivered, stress is heightened due to the gravity of patients’ conditions, long working hours, and the responsibility for making crucial decisions. Furthermore, the scarcity of advanced medical equipment and inadequate support services can further burden healthcare professionals at all levels. These unique stressors at each tier underscore the importance of targeted support and resources to mitigate stress and improve the overall well-being of healthcare professionals in Nigeria [88,89,90,91,92,93,94,95,96].

### 4.2. Health System Reform: A Pathway to Improve Stress Management among Healthcare Professionals

Health system reform in Nigeria has the potential to greatly improve stress management among healthcare professionals, as is emphasized in section four of [86]. For starters, thorough reform can address the long-standing concerns of insufficient staffing levels and onerous workloads, both of which contribute significantly to burnout and stress among healthcare professionals. Individual practitioners’ workloads can be reduced by expanding the number of healthcare professionals and imposing workload rules.

Additionally, investing in training and continuing education for healthcare professionals can boost their abilities and confidence, allowing them to deal with difficult situations more successfully. Access to psychological support and counseling services should be integrated into the healthcare system to help overburdened healthcare professionals with their emotional and mental well-being. Furthermore, implementing flexible working hours and offering adequate breaks might help to reduce fatigue and emotional exhaustion. Furthermore, the formation of support networks and peer-to-peer counseling platforms among healthcare professionals can build a sense of community and solidarity [7,19,86].

Furthermore, ensuring that healthcare providers receive fair and timely compensation can enhance morale and minimize financial stress. Involving healthcare professionals in decision-making and policy creation would empower them while also creating a more supportive work environment [19].

Public awareness campaigns and efforts can increase the appreciation and respect for healthcare professionals, lowering the stress caused by society’s misunderstandings or stigma.

By enacting these critical reforms, Nigeria’s health system would be able to provide a congenial and supportive environment for healthcare professionals, resulting in better stress management and, eventually, better patient care.

### 4.3. The Nigerian Experience: The Health Workforce Status

The Nigerian experience of the health workforce status differs significantly from the rest of the international healthcare industry, with a particular emphasis on the emigration of Nigerian healthcare professionals. Reports from the World Health Organization (WHO) highlight the country’s daunting task of delivering critical health services to cater to the basic humanitarian needs of over one million people, reflecting a pressing demand for healthcare resources and infrastructure [92]. One notable aspect is the persistent challenge of inadequate staffing levels, resulting in an overwhelming workload for healthcare professionals in Nigeria. This scarcity of healthcare workers has led to a surge in emigration, as healthcare professionals seek better opportunities and working conditions abroad. The ongoing issue of emigration further exacerbates the healthcare workforce shortage and compromises patient care within the country.

Compounding these challenges is the insufficient funding in Nigeria’s healthcare system, leading to disparities in salaries and limited resources for healthcare professionals [19,86]. The frustration caused by these circumstances has prompted healthcare professionals to resort to work strikes as a means to draw attention to their grievances and demand better working conditions and remuneration. The frequency of work strikes reflects the urgent need for comprehensive reforms in the health workforce status to retain skilled professionals and address the healthcare gaps in Nigeria.

Despite these difficulties, various initiatives and efforts are being made to improve the status of the healthcare workforce in the country. Implementing targeted policies that address emigration concerns, providing competitive remuneration packages, and improving working conditions can go a long way in retaining healthcare professionals and enhancing the overall healthcare system in Nigeria. Only through tailored and comprehensive solutions can the unique challenges faced by the healthcare workforce in Nigeria be effectively addressed, fostering a stronger and more resilient healthcare sector that can better serve the population’s needs [18,19,20,21].

### 4.4. What Has Been Newly Developed in the Field of Stress Management in General?

Following the pandemic, several countries have provided important guidelines and examples of practices for managing stress in the workplace. For example, in February of this year, the WHO presented a comprehensive guide on the consequences of stress and stressful situations, providing styles and symptoms, as well as teaching how to deal with stress, which can help our mental and physical well-being [97]. In addition, the Substance Abuse and Mental Health Services Administration (SAMHSA), an agency within the U.S. Department of Health and Human Services (HHS), has developed the guide Tips for Healthcare Professionals: Coping with Stress and Compassion Fatigue [98]. Also, the implementation of digital health interventions has recently been found to improve stress-coping behavior, and some Internet and app-based interventions were developed for the digital coaching of individual stress coping for healthcare workers [99].

## 5. Limitations

This study has some limitations, which might have influenced the results. The first is the low number and quality of studies from Nigeria. Thus, we could not determine whether there was a publication bias. Second, some studies did not mention in detail the outcomes, and so on, or gave too little detail, which obscured our methodological assessment.

## 6. Conclusions

This study discussed and presented the current literature on stress and stress management among healthcare workers in Nigeria in the literature, with the purview of identifying the main drivers of stress among health professionals and suggesting stress management promotion mechanisms among healthcare providers in the health sector. In this respect, stress in the healthcare industry has attracted a lot of attention due to its detrimental effects on both employees and patients. Health professionals tend to experience stress and burnout more frequently due to long job hours, caregiving responsibilities, and psychological contact with patients. The case of Nigerian health professionals is more glaring, considering the poor state of the healthcare facilities. This implies that Nigerian health professionals will have to deal with the stress they encounter when interacting with patients and the stress arising from poor health conditions. Taking all this into account, in order to reinforce Nigeria’s health system, prevent stress, and improve stress management, it is critical to address the persistent shortages and poor distribution of the health workforce. Nigeria needs to significantly increase investment in building up the health workforce to meet current and future needs. Strong measures are also needed to boost the training and recruitment of healthcare workers, as well as to improve their deployment and retention. Thus, this review also provides useful information for workplace health promotion to improve the workability of healthcare workers, with evidence-based content demonstrating the positive effects on stress management, in addition to content for the use of digital health in the context of healthcare work to improve effectiveness and adherence.

## 7. Recommendations

This paper proposes stress management strategies for healthcare professionals in Nigeria to address challenges such as insufficient medical equipment, poor work environments, inadequate salaries, and excessive workloads. It calls for a comprehensive health system reform to improve the well-being of healthcare professionals and enhance the quality of healthcare services. The strategies include mindfulness and resilience training, establishing support and counseling services, promoting work–life balance, and creating support networks. Health system reform should prioritize substantial funding for adequate medical equipment and resources, improve the work environment, address salary disparities, and implement workload regulations. By implementing these strategies, Nigeria can foster a healthier and more resilient healthcare workforce, ultimately leading to improved healthcare services. Collaborative efforts from policymakers, healthcare institutions, and other stakeholders are necessary to prioritize the well-being and productivity of healthcare professionals in the journey toward a more robust and equitable healthcare system.

## Figures and Tables

**Figure 1 healthcare-11-02815-f001:**
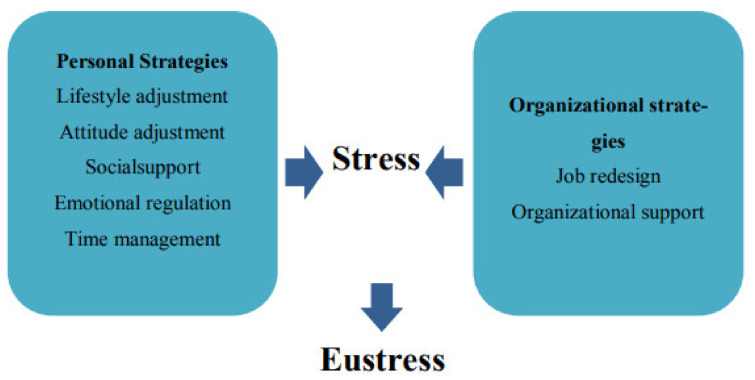
The stress management approach [81].

**Table 1 healthcare-11-02815-t001:** The whole set of search terms used in this review. However, the primary search keywords were stress AND management AND in healthcare organizations AND in selected health AND organization in AND Nigeria.

Search Words
“stress” OR “strain” OR “burnout” OR “tired” OR “mental stress” OR “physical stress”
AND
“Management” OR “organization” OR “sector” OR “leaders” OR “hospitals” OR “health” “organization”
AND
“Healthcare” OR “health sector “ OR “pharmacy” OR “health center” OR “primary healthcare organization”OR “secondary healthcare organization” OR “tertiary healthcare organization” OR “services” OR “health workers”OR “health professionals”
AND
“qualitative study” OR “interview” OR “discussion” OR “focus group” OR “field work” OR “qualitative research”OR “semi-structured” OR “unstructured” OR “structured” OR “informal” OR “in-depth” OR “face-to-face”
AND
“Nigeria” OR “Lagos” OR “Abuja” OR “Port Harcourt” OR “Ogun” OR “Kaduna” OR “Adamawa” OR “Gombe” OR “Taraba” OR “Yobe” OR “Ekiti” OR “Anambra” OR “Enugu” OR “Oyo” OR “Cross River” OR “Edo” OR “Delta” OR “Benue”

## Data Availability

Not applicable.

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
