# Peer review of "Stress Management in Healthcare Organizations: The Nigerian Context"

_healthcare, 2023, doi:10.3390/healthcare11212815_

Round 1

Reviewer 1 Report (New Reviewer)

In the abstract, the authors write "This narrative review seeks to provide information about the overall consequence of having over-stressed workers in healthcare and ascertain how it eventually draws back the advancement of healthcare ... in the Nigerian context."
One wonders, after analyzing the paper presented, whether the authors are entitled to draw such far-reaching conclusions. As a rule, narrative reviews are created by experts in a given field, who offer a recommendation based on their own solid knowledge and, above all, experience. At the same time, we may have to deal with misinterpretation of certain issues.
In addition, the definitions, types and strategies for coping with stress, which are too broadly presented in the work, have long been implemented in various health care units, as evidenced by numerous studies conducted in this environment.
The literature, consistent with the topic of the work, includes a variety of writing styles.

I believe that the work in its current form is not suitable for publication.

Author Response

Authors:

We appreciate and respect this reviewer's point of view. However, we do not understand what type of academic measurement R1 made to assume that our team is unable to carry out this narrative review. However, taking into account some criticisms, we included some texts and made some changes to the writing style.

Reviewer 2 Report (New Reviewer)

 The main objective of this article is to provide an overview of the current state of knowledge on stress management in the Nigerian healthcare sector.  The subject matter of the article is interesting and important in the practice of health services in Nigeria. It follows from the introduction that the article will be a review of the literature.

However, the authors should refine the structure of this article.

1.Introduction: points 2.1 to 2.5 in my opinion should be in the introduction.

Defining stress and its types is not something new that the authors will study. The introduction should not only clearly define the purpose of the article, but also the research questions.

2. Theoretical framework (point 3.4) should be before methodology and your research results. At this point you state what effect stress has on workers according to theory. and why it needs to be studied, including in the health service.

3. Methodology: In the methodology, the authors should use a more detailed description of the literature selection. For example, they could apply the Prisma model. (http://www.prisma-statement.org/PRISMAStatement/FlowDiagram)

The authors should describe how many articles were identified overall from this topic, how many were used and how many were excluded and why.

4. Research results: The results of the research should also show what new has been developed in the field of stress management in general, what can be applied in practice in health care, etc.

5. Conclusions are too short. Please state what the main conclusions of the literature review and what new trends in research the authors note.

Author Response

The main objective of this article is to provide an overview of the current state of knowledge on stress management in the Nigerian healthcare sector.  The subject matter of the article is interesting and important in the practice of health services in Nigeria. It follows from the introduction that the article will be a review of the literature. However, the authors should refine the structure of this article.

Authors: Thank you for the positive feedback regarding our study, as well as the constructive input.

1.Introduction: points 2.1 to 2.5 in my opinion should be in the introduction. Defining stress and its types is not something new that the authors will study. The introduction should not only clearly define the purpose of the article, but also the research questions.

Authors: Thanks you, we have add some text in the introduction to address this review comment.

  1. Theoretical framework (point 3.4) should be before methodology and your research results. At this point you state what effect stress has on workers according to theory. and why it needs to be studied, including in the health service.

Authors: Thanks you, done.

  1. Methodology: In the methodology, the authors should use a more detailed description of the literature selection. For example, they could apply the Prisma model. (http://www.prisma-statement.org/PRISMAStatement/FlowDiagram) The authors should describe how many articles were identified overall from this topic, how many were used and how many were excluded and why.

Authors: Thanks you, for the suggestion, but as we didn’t performed a systematic review, we have included  the whole set of search terms. In the other case, if we include the Prisma, we also should include the table with all papers results, (what it’s not the case, since we are providing an overall set of knowledge on this issue) , and in this case, will be need of 2 or 3 pages more, what will no fit in the numbers of available pages

  1. Research results: The results of the research should also show what new has been developed in the field of stress management in general, what can be applied in practice in health care, etc.

Authors: Thanks you, Done, please pg.13

  1. Conclusions are too short. Please state what the main conclusions of the literature review and what new trends in research the authors note.

Authors: Thanks you, Done,

Reviewer 3 Report (New Reviewer)

The work has been well written; only a few points should be addressed to make it more complete before publication. Therefore, addressing the following points is suggested:

The authors must specify the research gap that motivates the study in a paragraph in the introduction section.

Authors should add a section defining the practical and managerial implications of the findings reported in their study.

Authors should list any limitations of the study that make it difficult to generalize the results they report.

Author Response

The work has been well written; only a few points should be addressed to make it more complete before publication. Therefore, addressing the following points is suggested:

Authors: Thank you for the positive feedback regarding our study, as well as the constructive input.

The authors must specify the research gap that motivates the study in a paragraph in the introduction section.

Authors: Thanks you, done.

Authors should add a section defining the practical and managerial implications of the findings reported in their study

Authors: Thanks you, done.

Reviewer 4 Report (New Reviewer)

Thanks for the opportunity to read this study reviewing stress management in healthcare organisations. I enjoyed to read and review your paper. The feedback below elaborates on areas you may want to consider in the event of major revision.

Having said that, to my reading the paper lacks clear arguments, and is very much descriptive review. I would like to see a more elaborate findings on the review in the Nigeria, as well as a more elaborate context, definition, review approach, findings, discussion, and conclusion.

The title is unclear and imprecise. The title is focused on review, but most contexts are too general searching about stress management in healthcare organisations. This is invalid meaning, reflecting, and communicating to stress, especially healthcare organisations.

The abstract is too generally interpreted the purpose of study, findings and conclusion. I found the main findings are opposite in the key result contents.

The justification is poor and limited to review stress management in healthcare organisations. There are no reasons for examining gaps of knowledge, existing new model/framework, and objectives of the study.

The introduction is not related to relevant literature and provided objective of the study. However, the review misses literature that has employed stress management in healthcare organisations. The fact that theory may differ from review is something the authors mention in relation to the study's purpose and research objectives, but is not elaborated further in the analysis.

The method does not present any clear comprehensive review and synthesis of data, concepts or even sensitizing concepts with which to build an analysis. I found many point limitations in design, searching,  data extraction, and questions of conceptualization to review the validity of items. How many articles are included each idea? How many articles are excluded the study?

Should be clearly identified as follows:

1. Design

2. Search strategy

3. Eligibility criteria

4. Selection and data extraction

5. Data analysis

I am somewhat confused as to how the results are presented. More specifically, the key results are lacking an analysis of main reviews presented in the findings.

The way of reviewing the findings in fact does not differ from how the rest of the paper is written, in a very content-focused, review style, rather than bringing the different ‘Nigerian context’ of the findings rather than just summarizing the overall findings of the empirical review.

When explicitly formulating the contributions of the study, the findings of the current study are held up against the existing literature by mentioning different factors that seem more important for the improvement of theory than other studies have claimed.

Unfortunately, the lack of definition, theoretical, and review anchorage, which deprived the paper of a (theoretical) argument or direction, did not allow for the paper to invite scholarly engagement with the topic, as it remained describing all the ways in the spirit of summarizing results at a content level.

Should be provided as follows:

1. Practical implications

2. Theoretical implications

3. Policy implications

The conclusion should added and summarizes the content of this research, and it does little to emphasize the original and distinctive contribution of this research to advancing what we know about how to find new results. The conclusion is not summarized with the main findings.

Many grammatical errors. The communication is not in academic writing. The meaning is an invalid expression.

Author Response

Thanks for the opportunity to read this study reviewing stress management in healthcare organisations. I enjoyed to read and review your paper. The feedback below elaborates on areas you may want to consider in the event of major revision. Having said that, to my reading the paper lacks clear arguments, and is very much descriptive review. I would like to see a more elaborate findings on the review in the Nigeria, as well as a more elaborate context, definition, review approach, findings, discussion, and conclusion.

Authors: Thank you for the feedback regarding our study, as well as the input. I have add a couple to text and reviewed some topics to address this review concern.

The title is unclear and imprecise. The title is focused on review, but most contexts are too general searching about stress management in healthcare organisations. This is invalid meaning, reflecting, and communicating to stress, especially healthcare organisations.

Authors: Thank you for your comment, I have create the tittle according to our aims.

The abstract is too generally interpreted the purpose of study, findings and conclusion. I found the main findings are opposite in the key result contents.

Authors: Thank you. I have emended some points of the abstract to address this suggestion.

The justification is poor and limited to review stress management in healthcare organisations. There are no reasons for examining gaps of knowledge, existing new model/framework, and objectives of the study.

Authors: We appreciate this reviewer suggestion. We have included some corrections in the text to address this point.

The introduction is not related to relevant literature and provided objective of the study. However, the review misses literature that has employed stress management in healthcare organisations. The fact that theory may differ from review is something the authors mention in relation to the study's purpose and research objectives, but is not elaborated further in the analysis.

Authors. Thank you, some points was modified to the  introduction be  according the aims of the study.

The method does not present any clear comprehensive review and synthesis of data, concepts or even sensitizing concepts with which to build an analysis. I found many point limitations in design, searching,  data extraction, and questions of conceptualization to review the validity of items. How many articles are included each idea? How many articles are excluded the study?

Should be clearly identified as follows:

  1. Design
  2. Search strategy
  3. Eligibility criteria
  4. Selection and data extraction
  5. Data analysis

Authors.  Thank you, this point was already reported in the first review.

I am somewhat confused as to how the results are presented. More specifically, the key results are lacking an analysis of main reviews presented in the findings. The way of reviewing the findings in fact does not differ from how the rest of the paper is written, in a very content-focused, review style, rather than bringing the different ‘Nigerian context’ of the findings rather than just summarizing the overall findings of the empirical review. When explicitly formulating the contributions of the study, the findings of the current study are held up against the existing literature by mentioning different factors that seem more important for the improvement of theory than other studies have claimed.

Authors. Thank you for the suggestion. We share the reviewer’s opinion in some points and the authors try to address it.  .

Unfortunately, the lack of definition, theoretical, and review anchorage, which deprived the paper of a (theoretical) argument or direction, did not allow for the paper to invite scholarly engagement with the topic, as it remained describing all the ways in the spirit of summarizing results at a content level.

Should be provided as follows:

  1. Practical implications
  2. Theoretical implications
  3. Policy implications

The conclusion should added and summarizes the content of this research, and it does little to emphasize the original and distinctive contribution of this research to advancing what we know about how to find new results. The conclusion is not summarized with the main findings.

Please.

Authors. Thank you for the suggestion, please see the new revised manuscript.

Round 2

Reviewer 1 Report (New Reviewer)

The corrections made have actually changed the overall perception of the article. The results can serve as guidance for decision makers and not as the authors used the wording "recommendations" in the previous version.

References that are consistent with the topic of the paper contain different writing styles - this must be corrected.

Author Response

- The corrections made have actually changed the overall perception of the article. The results can serve as guidance for decision makers and not as the authors used the wording "recommendations" in the previous version.

Authors: Thank you for the positive feedback regarding our study, as well as the constructive input.

- References that are consistent with the topic of the paper contain different writing styles - this must be corrected.

Authors: Thank you

Reviewer 2 Report (New Reviewer)

The authors have attempted to improve the text. It is now more readable and understandable.

It is very good that the theoretical treatment of the problem has been emphasised.

The authors have written in the limitations of the point the practical and managerial implications (4.2), but without the content. It is essential to add them, as it is very important to apply the results to management practice.

In my opinion article should be published.

Author Response

- The authors have attempted to improve the text. It is now more readable and understandable. It is very good that the theoretical treatment of the problem has been emphasised.

Authors: Thank you for the positive feedback regarding our study, as well as the constructive input

- The authors have written in the limitations of the point the practical and managerial implications (4.2), but without the content. It is essential to add them, as it is very important to apply the results to management practice

. Authors: Thank you.  Done.

Reviewer 4 Report (New Reviewer)

Thank you for your revision. It is a better readability than previous version. All comments are revised and well-suited for publication in the Healthcare.

Moderate editing is required. This is some points missing meaning, grammar errors, and wrong sentence.

Author Response

- Thank you for your revision. It is a better readability than previous version. All comments are revised and well-suited for publication in the Healthcare.

Authors: Thank you for the positive feedback regarding our study.

- Comments on the Quality of English Language

- Moderate editing is required. This is some points missing meaning, grammar errors, and wrong sentence.

Authors: Thank you. We have revised the manuscript.

This manuscript is a resubmission of an earlier submission. The following is a list of the peer review reports and author responses from that submission.

Round 1

Reviewer 1 Report

Dear authors

Many thanks for submitting this manuscript for review and potential publication. Please see my comments below which I hope you will find constructive and helpful. 

Title - remove s from evidence. 

If a systematic review, the title needs to say same. 

Abstract - First sentence needs revising. 

You mention that this is a systematic review of the literature, however, I see no evidence of this throughout the paper. The abstract is not structured as per PRISMA 2020. There is no methodology to the paper. no mention of how the search was conducted, how they were screened and narrowed down, assessment of quality and bias, data synthesis etc. There is also no discussion section to this paper. To me, this is a discussion paper and I would  recommend that the authors recategorises this as such. As it stands the discussion within the paper is quite good but it is definitely not a systematic review. 

Keywords - 3 keywords is not sufficient, it should be at least 5. 

If of equal importance, then the keywords should be in alphabetical order. 

Introduction - you put numbers instead of the names of authors. Although the journal requires this, you could mention the authors followed by the number. So for example Christian et al. [91]. This is constant throughout the paper and needs to be rectified to improve readability. 

Be consistent with the language used. So is it health workers or healthcare workers or medical workers. Being consistent is key. 

Your sentences throughout are quite long winded. Please shorten them so that they are succinct and to the point. 

What do you mean by "on the foregoing..." 

Don't use e.g, etc in academic publications. use words like and so on. 

On page 3 you use & instead of and - rectify this as & should not be seen in academic writing. 

On page 4, you contradict yourself by saying high blood pressure and low blood pressure - please clarify. 

English language needs to be reviewed to ensure all sentences make sense and add something to the paper. 

At points in the paper, you make claims that are not necessary. See pg 6, section 2.9 first sentence. Please delete these sentences as they do not add anything to the paper. 

I do not see the value of either Figure 1 and 2. 

Theoretical framework section needs a lot more added to it. This is to be your standout part of the paper and yet it only receives 2 paragraphs. 

I hope you found this some way constructive. As mentioned earlier, I would suggest to the authors to review and resubmit as a discussion paper. I wish you well with your research. 

English language is moderately Ok but needs the support of someone reading who is fluent in English. 

Reviewer 2 Report

This paper lack cohesive structure and unclear primary argument, rationale or problem statement. The abstract is vague and not structured in a format that succinctly summarises the research questions, rationale, context, methods, theorrtical framework, results and future impact. I could not determine form the abstract what the writers intended in this paper and what their research design was. Is this a systematic review? 

The paper also requires a thorough review of English language usage for publication.

It is unclear what is new about this paper in terms of what is added to the literature surrounding psychological distress among healthcare workers other than to summarise the Nigerian context which only is presented at the very end of the paper. How does the Nigerian experience differ from the rest of the international healthcare industry. What is unique about the Nigerian experience? The authors go into detail of the various publicly funded tiers of the Nigerian Healthcare system but it is unclear if there are differences in these different settings that impact healthcare worker stress differently (how and why).  

The description of various forms of stress is too long and can be summarized in one paragraph or an image. This is well defined in current literature.

References should be identified by primary author--not just by number. 

The discussion lacks a "so what" and I am left at the end thinking what is main message and future implication of this paper.

There are significant grammatical errors that need to be reviewed.